# MiR-27a inhibits the growth and metastasis of multiple myeloma through regulating Th17/Treg balance

Weiguo Lu[1], Hui Huang[2,3], Zhanjie Xu[4], Shumin Xu[2,3], Kewei Zhao[1]*, Mingfeng Xiao[2,3]*

1 The Third Clinical Medical College, Guangzhou University of Chinese Medicine, Guangzhou, Guangdong, China, 2 The First Affiliated Hospital of Guangzhou University of Chinese Medicine, Guangzhou, China, 3 Guangdong Clinical Research Academy of Chinese Medicine, Guangzhou, China, 4 Guangzhou University of Chinese Medicine, Guangzhou, China

* zkw202420242024@163.com (KZ); xiaomingfeng2023@163.com (MX)

**Data Availability Statement:** The original contributions presented in the study are included in the Supplementary Material section of this manuscript. We have uploaded our original research data to the Harvard Dataverse. The data

## Abstract

### Background

The imbalance between T helper 17 (Th17) and T regulatory (Treg) cells plays a key role in the progression of multiple myeloma (MM).

### Methods

The gene expression profiles of MM were acquired and examined from the Gene Expression Omnibus (GEO) database (GSE72213). Our research involved experimental investigations conducted using the MOPC-MM mouse model. Dysregulation of Treg and Th17 cells was evaluated through flow cytometry, while the levels of inflammatory factors were measured using the enzyme-linked immunosorbent assay. Cell proliferation was gauged using the Cell Counting Kit-8 assay, and cell apoptosis was quantified via flow cytometry. Cell metastasis capabilities were determined by conducting transwell assays. To confirm the relationship between miR-27a and PI3K, a dual-luciferase reporter assay was employed. Finally, proteins associated with the PI3K/AKT/mTOR signaling pathway were assessed using western blotting.

### Results

MiR-27a exhibited reduced expression levels in MM. Moreover, it exerted control over the equilibrium of Th17 and Treg cells while reducing the expression of inflammatory mediators such as TGF-β1 and IL-10 in an *in vivo* setting. Elevated miR-27a levels led to the inhibition of cell viability, colony formation capacity, migratory and invasive traits in an *in vitro* context. The PI3K/AKT/mTOR signaling pathway was identified as a direct target of miR-27a and could reverse the effects induced by miR-27a in MM cells. Notably, PI3K was directly targeted by miR-27a.

can be accessed at the following link: https://doi.org/10.7910/DVN/BZATPL.

**Funding:** Administration of Traditional Chinese Medicine of Guangdong Province, China, Fund No.20241093; 2.Foundation of Guangdong Province, China, Fund No.C2023074 The recipients of the funding were responsible for the conception and design of this study.

## Conclusions

Our study revealed that miR-27a inhibited MM evolution by regulating the Th17/Treg balance. Inhibition of the PI3K/AKT/mTOR signaling pathway by miR-27a may play a potential mechanistic role.

## Introduction

Multiple myeloma (MM) ranks among the prevalent hematological malignances and is often associated with elevated immunoglobulin levels. The cytokines generated by plasma cells contribute significantly to localized damage and create a microenvironment conducive to the proliferation of malignant cells. Nevertheless, the underlying pathogenesis of MM remains unclear. The dysregulation of T lymphocyte subsets and the intricate cytokine network likely assume crucial roles in the development of MM [1, 2].

MicroRNAs (miRNAs) are a class of small non-coding RNAs, each consisting of 22 nucleotides in length. Their primary role involves the regulation of gene expression at the post-transcriptional level. Recent studies have highlighted the capacity of aberrantly expressed miRNAs to modulate processes such as cell growth, apoptosis, and tumorigenesis [3]. MiR-27a, located on chromosome 19 (19p13.1), is aberrantly expressed in various malignancies, including renal carcinoma, oral squamous cell carcinoma, and pancreatic cancer, among others [4]. MiR-27a is involved in the regulation of specific signaling pathways in multiple cancers, such as the AKT signaling pathway, Wnt/β-catenin signaling pathway, Ras/MEK/ERK signaling pathway, and TGF-β signaling pathway [5]. MiR-27a plays a diverse role in human cancer by impacting tumorigenesis, proliferation, apoptosis, invasion, migration, and angiogenesis through direct binding to the 3' untranslated region (3'UTR) of target messenger RNA (mRNA) [6]. In the context of multiple myeloma (MM), miR-27a has been reported to exert its biological influence by targeting Sprouty homolog 2 (SPRY2) [7].

Previous research has established that miR-27a regulates apoptosis in nucleus pulposus cells by targeting PI3K [8]. Another study has identified the pivotal role of PI3K/Akt signaling in balancing Th1/Th2 responses and has highlighted the critical nature of PI3K/Akt signaling in optimal Treg responses in pulmonary sarcoidosis [9]. Depending on the cytokines they produce and their functions, CD4+ T cells can be categorized into four subsets: T helper 1 (Th1), Th2, Th17, and CD4+ CD25+ T regulatory (Treg) cells [10]. Th1 lymphocytes are responsible for the secretion of interferon-gamma (IFN-γ), thus facilitating cell-mediated immune responses [11]. Conversely, Th2 cells are responsible for the production of IL-4, which suppresses the Th1 cell-mediated response. Th17 cells contribute to inflammation in the pathogenesis of various diseases by producing cytokines such as IL-17A, IL-6, and TNF-α [12, 13]. Treg cells inhibit effector T cell proliferation through secreting TGF-β and IL-10 to have immunomodulatory effects [14]. The imbalance between Th17 and Treg cells plays a key role in inflammatory and autoimmune diseases [15]. Recently, it was reported that Th17 cells may contribute to the development of MM and complications [16].

It's important to emphasize that miR-27a targets the 3'-UTR of the PI3K gene, thereby confirming a direct interaction between PI3K mRNA and miR-27a. Functional assessments have verified that miR-27a diminishes IL-1β-induced apoptosis and autophagy in MM. Conversely, inhibiting miR-27a leads to increased PI3K expression and activation of its downstream components, Akt and mTOR. These findings suggest the potential utility of miR-27a in the diagnosis and treatment of MM. To investigate whether the miR-27a PI3K/AKT/mTOR/Th17/Treg

axis contributes significantly to MM development, we conducted both *in vivo* and *in vitro* experiments to assess the proportion of Th subsets, serum levels of relevant cytokines, and the expression of specific transcription factors.

## Methods

### GEO database analysis

The GEO database (http://www.ncbi.nlm.nih.gov/geo) serves as a publicly available repository housing a multitude of datasets acquired through high-throughput sequencing and microarray analyses. These datasets originate from research institutions globally [17]. To pinpoint pertinent gene expression datasets, the keyword "MM" was employed during our search. We adhered to specific inclusion criteria, prioritizing expression profiles generated on the same sequencing platform with the most extensive sample size. Furthermore, we exclusively focused on datasets utilizing human test specimens. Finally, the microarray datasets [GSE72213 [18]] were downloaded from it (Affymetrix GPL570 platform, Affymetrix Human Genome U133 Plus 2.0 Array). The dataset includes 19 patients with three MM and three paired normal tissues. GEO2R (www.ncbi.nlm.nih.gov/geo/ge2r) is an online analysis tool developed using two R packages (GEOquery and Limma) [19]. We employed the GEOquery package to access the data and the Limma package for differential expression analysis. To identify the differentially expressed genes (DEGs) between the control and diseased groups, we used GEO2R to compare gene expression profiles. Any probe sets lacking corresponding gene symbols or genes represented by more than one probe set were excluded or averaged, respectively. The DEGs were limited to genes with adjusted P-value of <0.05 and |logFC (fold change) | values of ≥1.

### Tumor models

Experimental research was conducted using a mouse model of marrow-homing multiple myeloma (MOPC)-MM [20]. A total of 40 female 4-week-old BALB/c mice (Changzhou Cavens Experimental Animal Limited Company, Changzhou, China) were randomly assigned to either the MM model group or the normal group. The mice were anesthetized with 1.5% isoflurane, and efforts to alleviate suffering included monitoring the animals for signs of distress and providing appropriate analgesia when necessary. In the MM group, MM was induced by intravenously inoculating $2 \times 10^5$ MOPC cells (MOPC-315.Luc-GFP.BMP3 cells, tested negative for mycoplasma contamination) into the tail veins of the mice. The normal group consisted of healthy BALB/c mice. We compared the weight changes of the mice in different groups over a 2-week period. All animal care and experimental procedures were conducted in compliance with the guidelines established by the Institutional Animal Care and Use Committee of the First Affiliated Hospital of Guangzhou University of Chinese Medicine (GZTCMF1-20230501). The First Affiliated Hospital of Guangzhou University of Chinese Medicine granted ethical approval to conduct this study (Ethical Application Ref: K-2022-033).

### Quantitative real-time PCR

The mRNA expression of miRNA-27a was determined through real-time polymerase chain reaction (PCR). Total RNA was extracted using Trizol Reagent (Servicebio, China) according to the manufacturer's guidelines. The concentration and purity of RNA were assessed using an ultra-micro spectrophotometer (NanoDrop2000, Thermo, USA). Subsequently, cDNA was synthesized from the mRNA using the RevertAid First Strand cDNA Synthesis Kit (Thermo, USA). For cDNA synthesis, 1 µg of RNA was used with Evo M-MLV RT Premix for qPCR (Accurate Biotechnology Co., Ltd., Hunan, China). The primers sequences are shown in

**Table 1. Primers for qRT-PCR analysis.**

| Genes | | Sequences (5'→3') | Length |
|---|---|---|---|
| MiR-27a | Forward | GGCTAAGTTCCGCGTCG | 53bp |
| | Reverse | CAGTGCGTGTCGTGGAGT | |
| GAPDH | Forward | CACCCGCGAGTACAACCT | 138bp |
| | Reverse | CCCATACCCACCATCACACC | |

Table 1. A real-time PCR System (ABI, USA) was used to conduct qualitative real-time PCR, employing the FastStart Universal SYBR Green Master (Rox, Servicebio, China). The relative gene expression levels were calculated using the threshold cycle (CT), according to the $2^{-\Delta\Delta CT}$ method. Glyceraldehyde-3-phosphate dehydrogenase (GAPDH) was set as reference.

## Flow cytometry analysis and inflammation assessment

Flow cytometry was employed to investigate the frequencies of Th17 and Treg cells. The mice in the various groups were euthanized by cervical dislocation after retroorbital blood collection, and their bodies were immersed in 75% ethanol for 5 minutes. Spleen tissues were collected and placed in sterile PBS. The tissue was gently disrupted by pressing with the base of a sterile syringe, and the cell suspension was filtered through a 200-mesh cell screen. Red blood cells were lysed using a lysis buffer. Following cell counting, $1.0 \times 10^6$ cells were collected for flow antibody staining. The cell surface markers APC-CD25, PE-CD4, and BV421-FOXP3 (Biolegend, USA) were employed for the specific labeling of Treg cells, while FITC-CD3, PE-CD4, and BV510-IL-17A (Biolegend, USA) were used to identify Th17 cells. Subsequently, the stained cells were analyzed via flow cytometry. Furthermore, the inflammatory factors IL-2, IL-10, and TGF-β were quantified using enzyme-linked immunosorbent assay (ELISA) (Cusabio, China).

## Cell culture and transfection

Human RPMI-8226 (CRM-CCL-155) and NCI-H929 (CRL-9068) cells were procured from the American Type Culture Collection (ATCC; Manassas, VA, USA) and maintained in Roswell Park Memorial Institute 1640 Medium (RPMI1640; Gibco, Grand Island, NY, USA) supplemented with 10% fetal bovine serum (FBS, Gibco), 100 U/mL penicillin, and 100 mg/mL streptomycin in a 5% $CO_2$ humidified incubator at 37°C. MiR-27a mimic (miR-27a) and its corresponding negative control (NC), miR-27a inhibitor (anti-miR-27a), and its negative control (anti-NC) were obtained from GenePharma Co. Ltd. (Shanghai, China). The PI3K activator 740 Y-P and its inhibitor, 3-Methyladenine, were provided by MedChemExpress (Shanghai, China). The above-mentioned oligonucleotides (40 mM) or plasmids (2 μg) were transfected into RPMI-8226 and NCI-H929 cells using Lipofectamine 3000 (Life Technologies Corporation, Carlsbad, CA, USA). After 48 hours, the cells were harvested for subsequent assays.

## Cell viability and apoptosis assay

The Cell Counting Kit-8 (CCK-8, Beyotime) reagent was employed to assess the viability of MM RPMI-8226 and NCI-H929 cells according to the manufacturer's protocols. Cells were cultured in 0.5% serum for 12 h prior to the experiments. Cells were seeded in 96-well plates at a density of $5\times10^3$ cells per well and then transfected for 24, 48, and 72 hours. After the addition of 10 μL of CCK-8 reagent, the cells were further incubated for 1 hour at 37°C. The absorbance at 450 nm of each well was measured using a Microplate Reader (Bio-Rad, Hercules, CA, USA). To determine the apoptotic rate of RPMI-8226 and NCI-H929 cells, we employed

the Annexin V-fluorescein isothiocyanate (FITC) Apoptosis Detection Kit (Beyotime). A total of $2 \times 10^5$ cells were seeded in a 6-well plate and incubated for 24 hours. Subsequently, 10 μL of Annexin V-FITC and propidium iodide (PI) solution were added. After a 15-minute incubation, apoptotic cells were analyzed using a flow cytometer (Beckman Coulter, Fullerton, CA, USA), and the apoptotic rate was determined by assessing the proportion of cells that were Annexin V+/PI-.

## Transswell assay

The invasion and migration capacities of RPMI-8226 and NCI-H929 cells were assessed using a Transwell assay with Transwell chambers (Corning Inc., Corning, NY, USA). To evaluate invasion ability, $4 \times 10^4$ cells in serum-free medium were placed into the upper chambers coated with Matrigel (Corning Inc.). For migration ability assessment, $1 \times 10^4$ cells in serum-free medium were seeded into the upper chambers. The lower chambers were filled with medium containing 10% FBS. After 24 hours of incubation, the migrated or invasive cells were stained with 0.1% crystal violet and counted under a light microscope (at 200× magnification).

## Dual-luciferase reporter assay

Wild-type luciferase reporter vectors (PI3K-wt) and mutant-type ones (PI3K-mut) were established by inserting corresponding sequences into luciferase reporter vector pGL3 (Promega, Madison, WI, USA). The constructed vectors and miR-27a or NC were cotransfected into RPMI-8226 and NCI-H929 cells, and after 2 h, luciferase activity was evaluated using the Dual-Lucy Assay Kit (Solarbio).

## Western blot

First, RPMI-8226 and NCI-H929 cells were subjected to protein isolation using a protein extraction kit (Solarbio, Beijing, China). After quantification with a bicinchoninic acid protein assay kit (Beyotime), 30 μg protein samples were loaded on a 12% sodium dodecyl sulfate–polyacrylamide gel and then transferred onto polyvinylidene difluoride membranes (Pall Corporation, East Hills, NY, USA). The membranes were blocked with skim milk, followed by incubation with specific primary antibodies: PI3K antibody (1:2500, ab227204, Abcam), p-PI3K antibody (1:2500, ab182651, Abcam), mTOR antibody (1:1000, ab2732, Abcam), p-mTOR antibody (1:2000, ab109268, Abcam), Akt antibody (1:2000, ab8805, Abcam), p-Akt antibody (1:500, ab38449, Abcam), and β-actin (1:2000, ab8227, Abcam). Subsequently, the membranes were incubated with the secondary antibodies conjugated with Amersham ECL peroxidase: goat anti-rabbit IgG (1:5,000) or goat anti-mouse IgG (1:5,000) at room temperature for an additional 60 minutes. The immunoreactivity was measured using a Super Signal West Femto Maximum Sensitivity Substrate Kit (Thermo) on a C-DiGit Blot Scanner.

## Statistical analysis

For data analysis and figure generation, IBM SPSS Statistics for Windows, version 20 (IBM Corp., Armonk, N.Y., USA) and GraphPad Prism, version 8.0.1 (GraphPad Software, San Diego, CA, USA) were employed, respectively. All samples were measured in triplicate and the average values were calculated. The data are presented as the means ± standard deviation (SD). The normality and homogeneity of the results were assessed using the Shapiro-Wilk test. The experimental data were analyzed using one-way analysis of variance (ANOVA) and two-way ANOVA, followed by Tukey's multiple comparisons test. Statistical significance was determined at $p < 0.05$.

# Results

## Identification of DEGs

After normalizing the microarray data, differentially expressed genes (DEGs) were identified from the GSE72213 dataset. Through an intersection analysis, it was observed that the expression of miR-27a was significantly higher in normal samples compared to that in MM tissues. Furthermore, when compared to healthy controls, the expression of PI3K, AKT, and mTOR was significantly elevated in MM samples (Fig 1A and 1B). Consequently, miR-27a was selected for further validation and in-depth investigation.

## Verification of the relationship between miR-27a and Th17/Treg balance *in vivo*

As depicted in Fig 2A, MOPC-MM mice exhibited a significant decrease in weight gain within a 2-week period compared to that of the control group (Control 2 Weeks: 18.38±1.29 g vs. MOPC-MM 2 Weeks: 17.09±1.14 g, p <0.001). The levels of miR-27a were assessed by qRT-PCR. The specific miR-27a expression levels were notably higher in the spleens of the control group compared to those in the MOPC-MM group (Fig 2B).

Fig 2C and 2D illustrate an analysis of the populations of CD25+ T cells expressing Foxp3, showing that the number of Treg cells in the spleens of the MOPC-MM group increased by 3.27-fold compared to that of the control group (Control: 1.78±0.22% vs. MOPC-MM: 5.83 ±0.12%, p = 0.004). Conversely, the number of Th17 cells in the spleens of the MOPC-MM group decreased by 5.21-fold compared to that of the control group (Control: 5.68±0.49% vs. MOPC-MM: 1.09±0.17%, p = 0.02). Results of ELISA demonstrated a marked increase in IL-2 levels in the spleens of the control group compared to those in the MOPC-MM group. However, the levels of IL-10 and TGF-β were reduced in the control group relative to those in the MOPC-MM group (Fig 2E). These findings suggested that the MOPC-MM model experienced a severe inflammatory state owing to an imbalance in the Th17/Treg ratio.

## Low expression of miR-27a promoted proliferation and inhibited apoptosis as well as invasion of MM cells

To elucidate the impact of miR-27a on the cellular behavior of MM *in vitro*, we transfected RPMI-8226 and NCI-H929 cells with the miR-27a mimic and miR-27a inhibitor, while cells

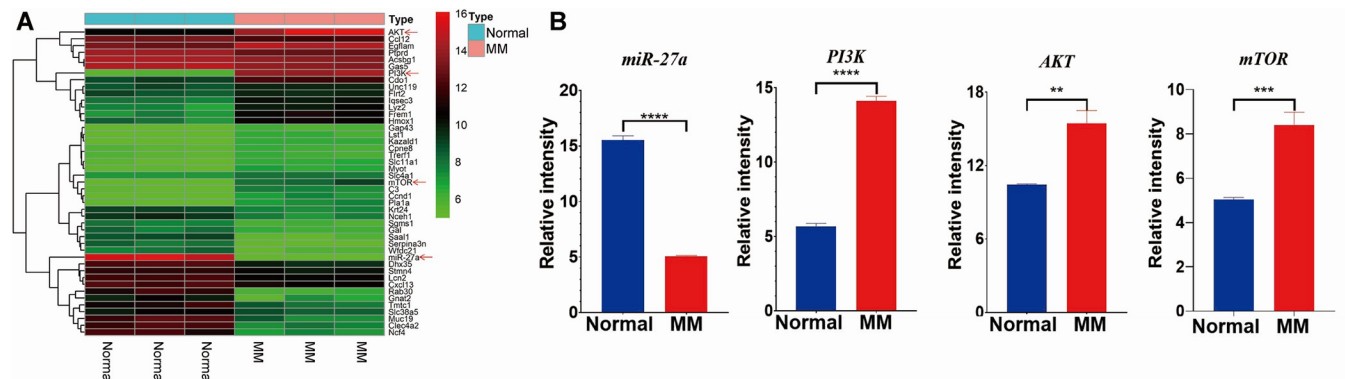

**Fig 1. Microarray data.** (A) Different expression levels of a set of DEGs in GSE72213. Green, low expression levels; red, high expression levels. (B) Histograms displaying the different expressions of miR-27a, PI3K, AKT, and mTOR, as detected by the microarray analysis of MM samples relative to healthy controls. *P < 0.05, **P < 0.01, ***P < 0.001, ***P < 0.0001.

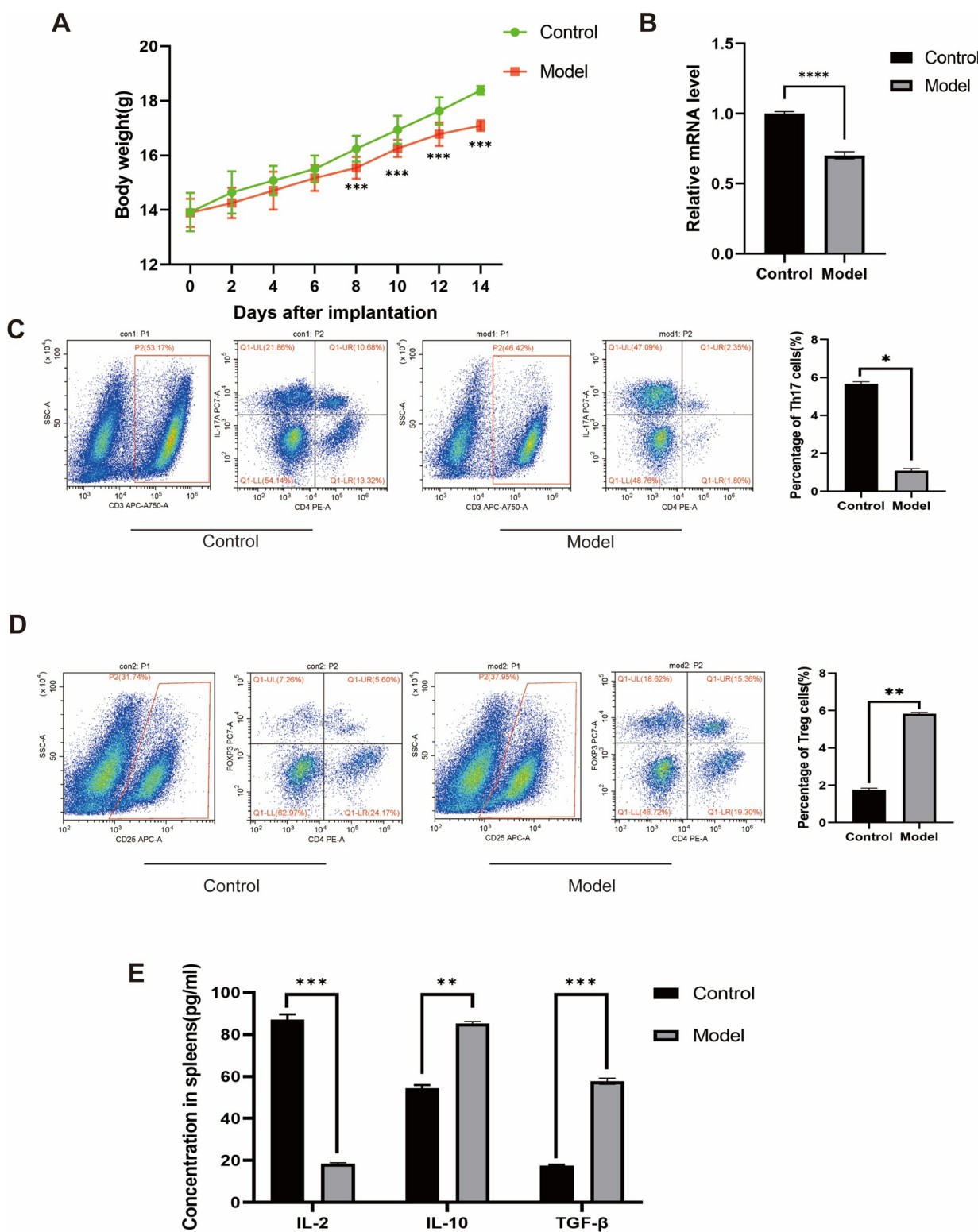

**Fig 2. Relationship between MiR-27a and MM *in vivo*.** (A) Time course changes of body weight. (B) The relative expression of miR-27a in spleens of MOPC-MM was analyzed by qRT-PCR assay, and normalization to the levels of GAPDH. (C) Flow cytometry of the Th17 cells in the MOPC-MM's spleens. (D) Flow cytometry of Tregs in the MOPC-MM's spleens. (E) The protein levels of IL-2, IL-10 and TGF-β in the spleens of each group. All the values are the means ± S.D. *P < 0.05, **P < 0.01, ***P < 0.001, ****P < 0.0001.

transfected with miR-27a-NC served as controls. The CCK-8 assay revealed an inhibitory effect of miR-27a on the viability of RPMI-8226 and NCI-H929 cells (Fig 3A). Our data demonstrate that introducing the miR-27a mimic for 48h and 72h inhibited the proliferation of both RPMI-8226 and NCI-H929 cells in a dose-dependent manner (P <0.05). Flow cytometry analysis indicated that the overexpression of miR-27a promoted cell apoptosis and led to cell cycle arrest in the G1 phase when compared to the control and anti-miR-27a groups (Fig 3B). Furthermore, the introduction of miR-27a mimic reduced the number of migrated and invaded RPMI-8226 and NCI-H929 cells (Fig 3C). In summary, the introduction of miR-27a suppressed proliferation and metastasis but enhanced apoptosis in MM RPMI-8226 and NCI-H929 cells.

## MiR-27a alleviated the severity of inflammatory state of MM cells

The concentrations of serum cytokines, including IL-2, IL-10, and TGF-β, were assessed using ELISA in various groups of RPMI-8226 and NCI-H929 cells. As depicted in Fig 4, the levels of IL-2 were notably elevated in the miR-27a mimic group, while the levels of IL-10 and TGF-β were lower than those in the anti-miR-27a group. Additionally, the levels of IL-10 were significantly elevated in the anti-miR-27a group.

## MiR-27a directly targeted PI3K/AKT/mTOR signaling

Subsequently, our aim was to elucidate the impact of miR-27a on MM development through the PI3K/AKT/mTOR signaling pathway. The results from the dual-luciferase reporter assay indicated that the miR-27a mimic (with control miRNA transfection as a reference) significantly suppressed the expression of the reporter gene containing the 3'UTR of PI3K, while it had no effect on the expression of the reporter gene containing the mutated variant of the 3'UTR of PI3K (Fig 5A). This observation underscores the direct binding and inhibitory effect of miR-27a on PI3K expression. Subsequent western blot analysis revealed a reduction in the levels of p-PI3K, p-AKT, and p-mTOR in RPMI-8226 cells transfected with the MiR-27a mimic (Fig 5B).

## Upregulation of miR-27a counteracted the effects of PI3K/AKT/mTOR signaling on the cellular behaviors of MM cells

Next, we conducted a rescue experiment to investigate how miR-27a modulates the cellular behaviors of RPMI-8226 and NCI-H929 cells. The enhancing effects of PI3K/AKT/mTOR signaling on cell viability and colony formation ability in RPMI-8226 cells and NCI-H929 were attenuated by the presence of the miR-27a mimic. Furthermore, the introduction of miR-27a reinforced apoptosis and induced cell cycle arrest in the G1 phase, as revealed by flow cytometry (Fig 6A). Additionally, the inclusion of miR-27a inhibited the metastatic potential of both two MM cell lines induced by PI3K/AKT/mTOR signaling (as illustrated in Fig 6B). As anticipated, the miR-27a mimic-mediated decrease in p-PI3K, p-AKT, and p-mTOR levels in RPMI-8226 and NCI-H929 cells was reversed upon PI3K activation (Fig 6C). In summary, the overexpression of miR-27a inhibited the progression of MM by suppressing the expression of PI3K/AKT/mTOR signaling.

## Discussion

It is widely accepted that MM and signaling pathways are implicated in the occurrence and development of MM, including miRNAs [21]. In the present study, the inhibitory role of miR-27a was established in MM both *in vitro* and *in vivo*. Furthermore, we provided the initial

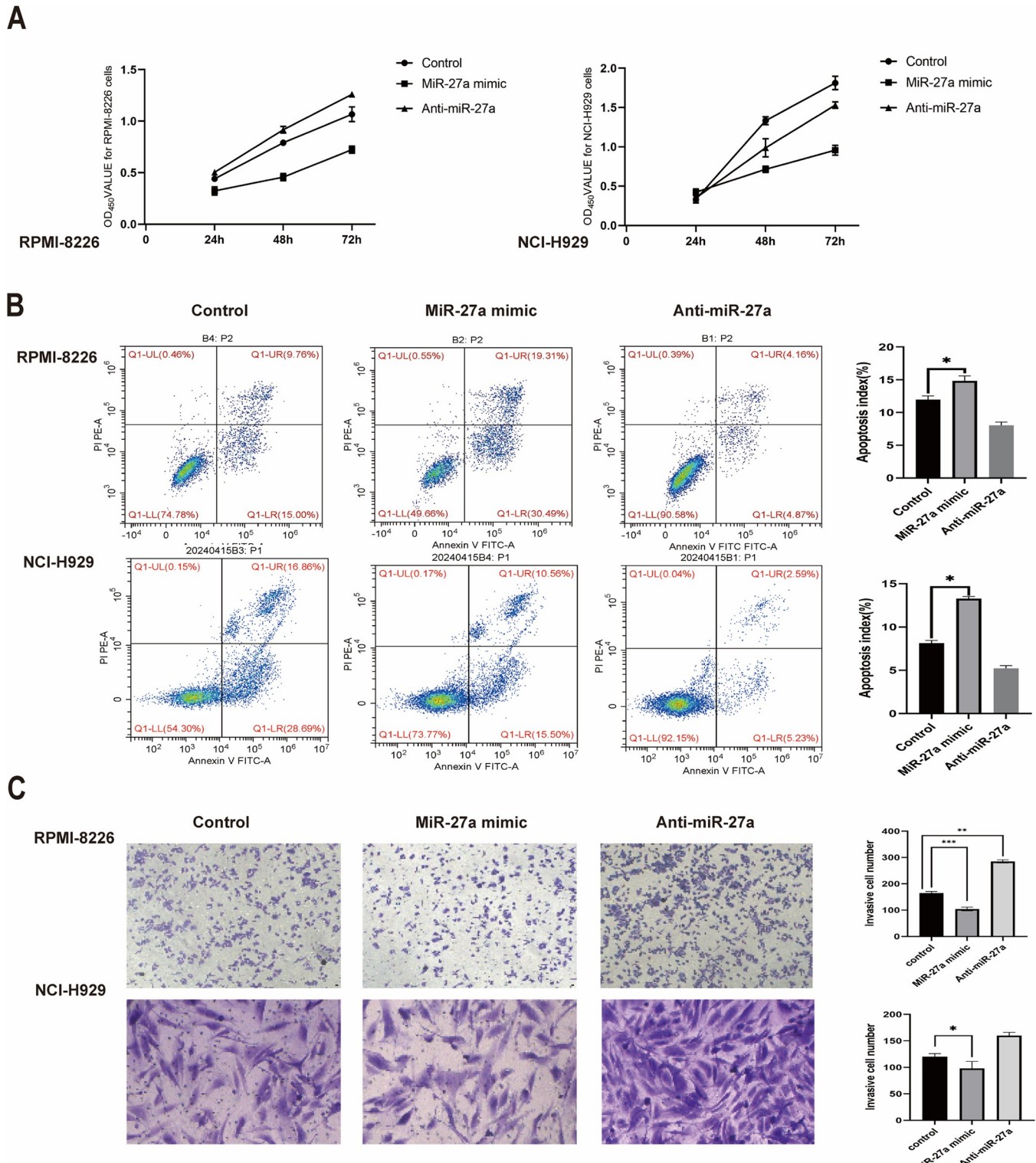

**Fig 3. Overexpression of miR-27a inhibited proliferation and invasion, but promoted apoptosis of MM cells.** (A) Cell viability of two transfected cell lines was analyzed using the CCK-8 assay. (B) The apoptosis rate and cell cycle distribution were tested using the flow cytometry assay. (C) The migration and invasion abilities of transfected cells were examined via the transwell assay. $^*P < 0.05$, $^{**}P < 0.01$, $^{***}P < 0.001$.

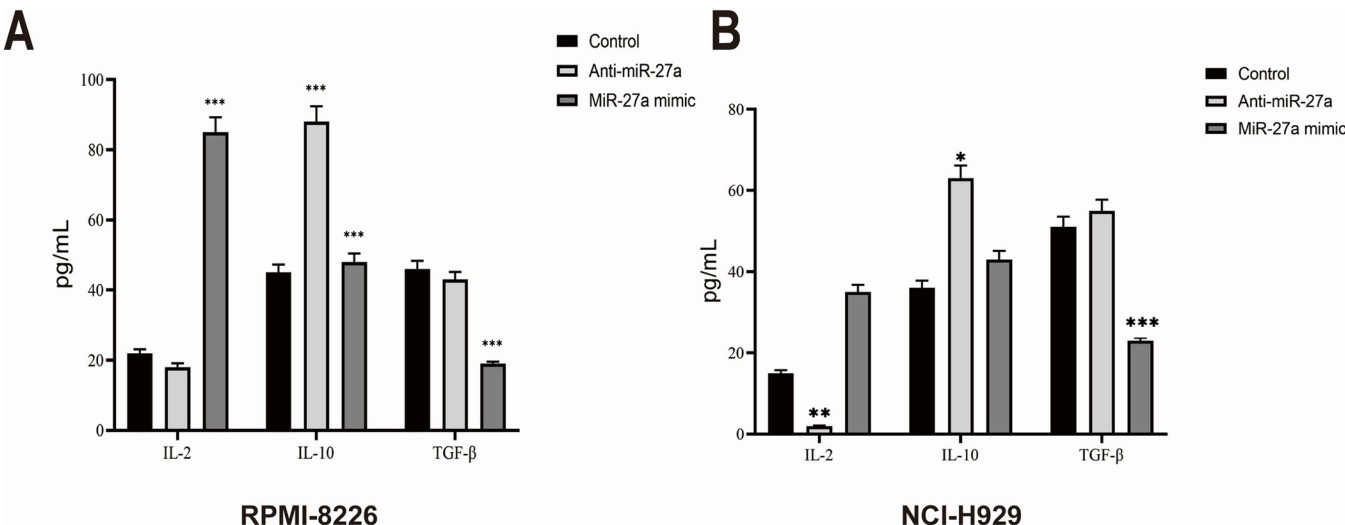

**Fig 4. MiR-27a alleviated the severity of inflammatory state of MM cells.** (A) The protein levels of IL-2, IL-10 and TGF-β in the RPMI-8226 cells of each group. (B) The protein levels of IL-2, IL-10, and TGF-β in the NCI-H929 cells of each group. All values are expressed as the means ± S.D. *P < 0.05, **P < 0.01, ***P < 0.001.

validation of the regulatory impact of the MiR-27a/PI3K/AKT/mTOR axis on the modulation of Th17/Treg balance in MM.

miRNAs are small non-coding RNAs that have been highly conserved throughout evolution and can modulate gene expression at the translational level [22]. They have been implicated in various cellular processes and exert downstream regulatory effects on various genes. Additionally, they exhibit stability in biological fluids, such as urine, blood, and saliva, facilitating sample collection and advancing miRNA research across multiple domains [23]. In tumor tissues, certain miRNAs demonstrate reduced expression levels, acting as suppressors of tumorigenesis, whereas others exhibit increased expression, promoting tumor advancement [24]. Consequently, miRNAs can serve as potential diagnostic markers and prognostic indicators for tumor detection and evaluation.

Previous research has indicated that miR-27a can enhance autophagy and apoptosis in IL-1β-treated articular chondrocytes in osteoarthritis [25]. Additionally, miR-27a has been shown to increase cisplatin sensitivity in hepatocellular carcinoma cells by inhibiting the PI3K/Akt pathway [26]. These findings underscore the dual functionality of miR-27a in human malignancies, potentially influenced by variations in the tumor microenvironment.

In this basic bioinformatics study, we observed that miR-27a is down-regulated in MM tissue, serving as a potential indicator of favorable prognosis for MM patients. Our *in vitro* and *in vivo* assays consistently demonstrated that miR-27a functions as a tumor suppressor gene in MM, inhibiting cell viability, inducing apoptosis, and affecting the cell cycle. However, it remained unclear whether miR-27a played a role in regulating the balance between Th17 and Treg cells. Significantly, for the first time, we report that the overexpression of miR-27a markedly reduces the proportion of Treg cells while enhancing Th17 cells *in vivo*. This finding suggests that miR-27a plays a crucial role in regulating the balance between Th17 and Treg cells. Additionally, we have preliminary evidence indicating that miR-27a achieves this by targeting the PI3K/AKT/mTOR signaling pathway. Our data underscore the potential of miR-27a as a novel molecular target for MM chemotherapy.

In our *in vitro* assays, we demonstrated that the overexpression of miR-27a inhibits MM cell viability, promotes apoptosis, and induces cell cycle arrest in the G0/G1 phase. This

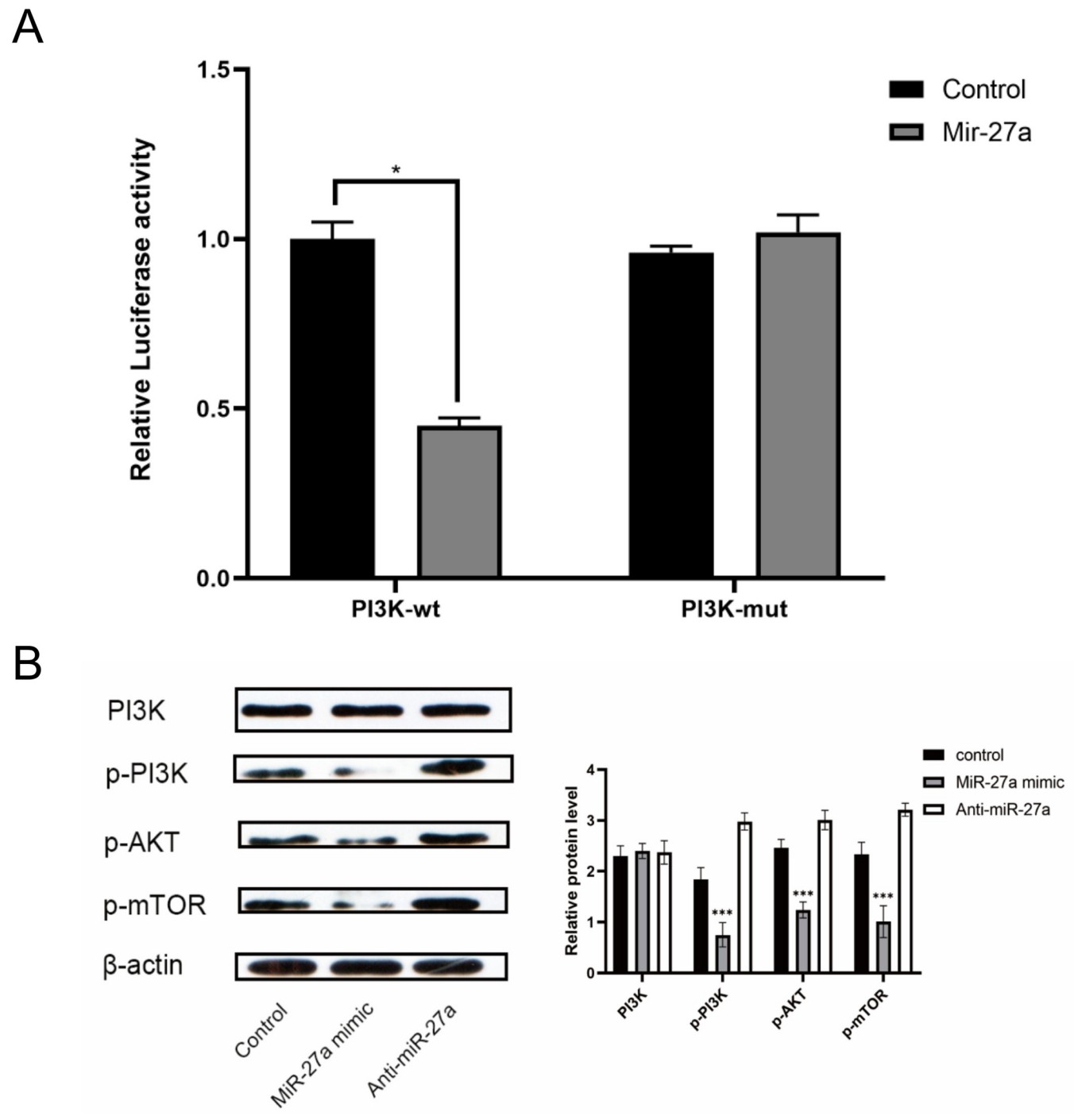

**Fig 5. MiR-27a could target PI3K/AKT/mTOR signaling in MM.** (A) The luciferase activity in the PI3K-wt group and PI3K-mut group of RPMI-8226 cells was determined via the dual-luciferase reporter assay. (B) The expression of PI3K, p-PI3K, p-AKT, and p-mTOR proteins in RPMI-8226 cells was measured via western blot assay. *P < 0.05, **P < 0.01, ***P < 0.001.

highlights the pivotal role of miR-27a in MM tumorigenesis. However, it's worth noting that the functions of miR-27a can vary in different types of cancer. For instance, in nasopharyngeal carcinoma, Li et al. found that miR-27a promotes cell proliferation, enhances migration, and facilitates invasion by targeting the Mapk10 protein [27]. Additionally, Mertens-Talcott et al.

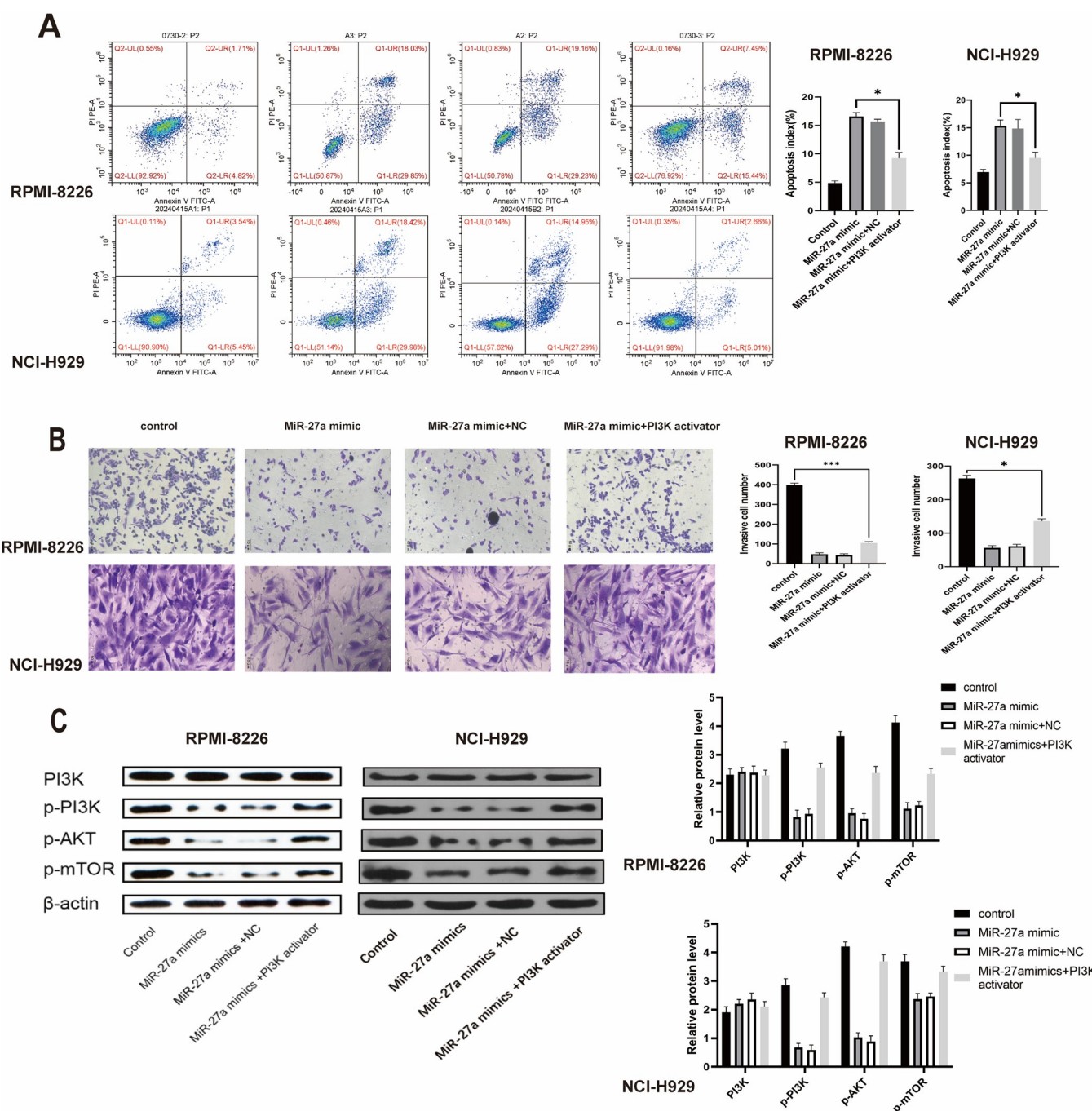

**Fig 6. Upregulation of miR-27a counteracted the effects of PI3K/AKT/mTOR signaling on RPMI-8226 and NCI-H929 cells.** (A) The apoptosis rate and cell cycle distribution were tested via the flow cytometry assay. (B) The migration and invasion abilities of transfected cells were examined using the transwell assay. (C) The expression of PI3K, p-PI3K, p-AKT, and p-mROT proteins in RPMI-8226 and NCI-H929 cells was measured via western blot assay. *P < 0.05, **P < 0.01, ***P < 0.001.

observed an oncogenic role of miR-27a in breast cancer, where it increased the percentage of breast cancer cells in the G2-M phase by targeting the Myt-1 gene [28]. These examples illustrate that miR-27a can play distinct roles in different cancer types.

As we have learned, the imbalance between Th17 and Treg cells is a key factor in many diseases [29]. Although there is conflicting evidence regarding the role of Th17 cells in cancer, excessive inflammation caused by Th17 cells or an overactive immunosuppression by Treg cells may contribute to carcinogenesis [30, 31]. The effect of Th17 cells on cancer development depends on the tumor's phenotype, while Treg cells are typically linked to tumor progression and reduced survival rates in cancer patients [32]. In tumors, the expression of GARP has been shown to enhance active TGF-β, promoting Treg induction within the cancer microenvironment and thus inhibiting immune responses [33]. The presence of Treg cells in the tumor microenvironment is associated with advanced malignancy stages, invasive growth, and poorer prognoses [32]. In our study, we observed a significantly higher percentage of Treg cells and a lower percentage of Th17 cells in MM samples compared to healthy controls. Following the assessment of relevant inflammatory factors, we discovered that the increase in Treg cells was associated with higher levels of IL-10 and TGF-β, and it was also closely correlated with miR-27a expression. In addition, the inflammatory effects of IL-2, IL-10, and TGF-β have significant implications in MM. IL-2, known for its role in T-cell proliferation and activation, appears to be suppressed in MM, contributing to impaired immune responses. Conversely, IL-10, with its anti-inflammatory properties, may create a microenvironment conducive to MM progression by dampening immune surveillance and promoting tumor cell survival [34]. Additionally, TGF-β, a multifunctional cytokine, plays a dual role in MM pathogenesis. Although it exerts tumor-suppressive effects by inhibiting cell proliferation and inducing apoptosis, its immunosuppressive actions can foster MM growth and evasion of immune surveillance [35]. Thus, the interplay between these inflammatory mediators underscores their intricate involvement in MM pathophysiology and suggests potential therapeutic targets.

Based on these prior studies, PI3K and its downstream components, namely Akt and mTOR, have been associated with multiple biological processes [36, 37]. In particular, phosphorylated Akt influences various genes, especially pro-apoptotic factors such as Bcl-xL, Bcl2, Bax, VEGF, and MMPs [38–40]. In our study, we observed that miR-27a impacts the development of MM through the PI3K/AKT/mTOR signaling pathway. Notably, the reduction in p-PI3K, p-AKT, and p-mTOR levels induced by miR-27a mimic was reversed upon PI3K activation. Although our research briefly addressed the effect of tumor microenvironment variations on miR-27a 's dual functionality, a more comprehensive examination of potential limitations, particularly the specific conditions dictating miR-27a 's diverse roles, is warranted. Investigating these aspects would offer a more nuanced interpretation of our findings and shed light on the context-dependent nature of miR-27a 's functions. Despite these limitations, our findings suggest that miR-27a likely plays a significant role in suppressing MM progression by inhibiting the expression of the PI3K/AKT/mTOR signaling pathway.

## Conclusions

To conclude, miR-27a functions as a tumor suppressor gene in MM. Significantly, we verified the inhibition of MM growth and metastasis by miR-27a through regulation of the Th17/Treg balance. Additionally, we acknowledge the potential mechanistic role of the PI3K/AKT/mTOR signaling pathway in this process. Our findings emphasize the potential of miR-27a as a molecular target in MM chemotherapy strategies.

## Supporting information

**S1 File. This file contains the original PCR data used in the study, including the amplification curves, threshold cycles (Ct values), and gel electrophoresis images for the target**

**genes.**
(ZIP)

**S2 File. This file includes the original Western blot data, featuring raw images of the blots for the target proteins, along with corresponding molecular weight markers.** Detailed information on the antibody used, sample preparation, and exposure times is also provided.
(ZIP)

**S3 File. This file contains the original flow cytometry images, including dot plots and histograms for the analyzed samples.**
(ZIP)

**S4 File. This file includes the original images from the transwell assays, showcasing the migratory and invasive capacities of the cell populations under various experimental conditions.**
(ZIP)

## Acknowledgments

We would like to thank Editage for providing language editing for this manuscript.

## Author Contributions

**Conceptualization:** Weiguo Lu, Mingfeng Xiao.

**Data curation:** Weiguo Lu, Hui Huang.

**Methodology:** Shumin Xu.

**Supervision:** Kewei Zhao.

**Writing – original draft:** Zhanjie Xu.

**Writing – review & editing:** Kewei Zhao, Mingfeng Xiao.

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
