## [Decision Letter · Decision Letter 0]

16 Nov 2023

PONE-D-23-34574MiR-27a inhibits the growth and metastasis of multiple myeloma through regulating Th17/Treg balance via PI3K/AKT/mTOR signalingPLOS ONE

Dear Dr. Xiao,

Thank you for submitting your manuscript to PLOS ONE. After careful consideration, we feel that it has merit but does not fully meet PLOS ONE’s publication criteria as it currently stands. Therefore, we invite you to submit a revised version of the manuscript that addresses the points raised during the review process. This manuscript was carefully reviewed by 2 experts. Although the results presented in this study are of potential importance and interest, the reviewers raised a number of issues which need to be addressed before acceptance. Please respond to each of the reviewer comments.

We look forward to receiving your revised manuscript.

Kind regards,

Hiromu Suzuki, M.D., Ph.D.

Academic Editor

PLOS ONE

Journal Requirements:

2. To comply with PLOS ONE submissions requirements, in your Methods section, please provide additional information regarding the experiments involving animals and ensure you have included details on (1)  methods of anesthesia and/or analgesia, and (2) efforts to alleviate suffering

- https://doi.org/10.1007/s10753-014-9980-4

In your revision ensure you cite all your sources (including your own works), and quote or rephrase any duplicated text outside the methods section. Further consideration is dependent on these concerns being addressed.

4. Thank you for stating the following financial disclosure: "1.Administration of Traditional Chinese Medicine of Guangdong Province, China，Fund No.20241093；2.Foundation of Guangdong Province, China，Fund No.C2023074"

Additional Editor Comments:

This manuscript was carefully reviewed by 2 experts. Although the results presented in this study are of potential importance and interest, the reviewers raised a number of issues which need to be addressed before acceptance. Please respond to each of the reviewer comments.

Reviewers' comments:

Reviewer's Responses to Questions

**Comments to the Author**

1. Is the manuscript technically sound, and do the data support the conclusions?

Reviewer #1: Partly

Reviewer #2: Partly

2. Has the statistical analysis been performed appropriately and rigorously? 

Reviewer #1: Yes

Reviewer #2: No

3. Have the authors made all data underlying the findings in their manuscript fully available?

Reviewer #1: Yes

Reviewer #2: Yes

4. Is the manuscript presented in an intelligible fashion and written in standard English?

Reviewer #1: Yes

Reviewer #2: Yes

5. Review Comments to the Author

Reviewer #1: Title: MiR-27a inhibits the growth and metastasis of multiple myeloma through regulating

Th17/Treg balance via PI3K/AKT/mTOR signaling

This study investigates the role of MiR-27a in regulating the balance between T helper 17 (Th17) and T regulatory (Treg) cells in multiple myeloma (MM). The research reveals that MiR-27a has reduced expression in MM and suggests an influence on the equilibrium of Th17 and Treg cells, reducing the expression of inflammatory mediators. Elevated MiR-27a levels inhibit cell viability, migration, and invasiveness, induce apoptosis, and arrest cell cycle progression in MM cells by targeting the PI3K/AKT/mTOR signaling pathway. This suggests that MiR-27a may play a role in inhibiting MM progression by modulating this pathway.

Comments:

1. Text on line 121 has a different font color.

2. Was DNA purity tested by running a gel too?

3. Where were primers purchased from?

4. Were cells serum-starved prior to viability and apoptosis assays?

5. GEO database analysis lacks details of how analysis was conducted, there is not sufficient detail for the study to be replicated.

6. Just to clarify, only 3 normal and 3 MM datasets were selected? This is what is represented in the clusterogram.

7. Figure 2B, perhaps include MiR-27a in the y axis label.

8. Figure 3C legend does not fully describe what is seen. This is also not well described in text.

9. I am not fully convinced by this statement: initial validation of the regulatory impact of the MiR-27a/PI3K/AKT/mTOR axis on the modulation of Th17/Treg balance in MM. The study does show the dual functionality of MiR-27A but does not clearly validate the regulatory impact of the MiR-27a/PI3K/AKT/mTOR axis on the modulation of Th17/Treg balance in MM. Although this study showed that miR-27a impacts the development of MM through the PI3K/AKT/mTOR signaling pathway, this cannot infer its modulation of the Th17/Treg balance. This is a major concern as it is the basis of the study and a conclusion that is drawn. The study does hold significant merit in its findings regarding the PI3K/AKT/mTOR pathway and MiR-27a.

Reviewer #2: The manuscript "MiR-27a inhibits the growth and metastasis of multiple myeloma through regulating Th17/Treg balance via PI3K/AKT/mTOR signaling" provides a good insight into miRNA based regulation of multiple myeloma. However, addressing the following comments can improve the manuscript.

1. In the abstract the conclusion is presented very vaguely, please improve the sentence structure to make it more scientific and clear to understand.

2. Introduction is satisfactory, the last lines on page 4 (84to 87) needs clarity of the aim of studies.

3. Methods are presented well, however certain details are missing. Please explain what is meant by control in case of cells are they untransfected cells or mock transfected? The information about the designing of miR-27a mimic and anti-miR-27a is missing. How the mutant of PI3K was constructed? Experiments replication number is not mentioned. The ethical permissions for animal studies should also be mentioned in methods section.

4. In the Results the statistical analysis in all the figures is not clear, the authors should double check all the analysis as at some points e.g in fig 2 it says **** but in the figure caption it is not mentioned. Also in the mRNA analysis the values on y-axis and stats doesnot coordinate. Please double check.

Figure 3c shows unequal contrast of the images

Fig 4 is not visible in the document

The authors may show some other in-vitro metastasis assays such as ECM degradation to support their hypothesis with appropriate controls

5. The discussion section needs improvement in corelating the results to the conclusion made. The authors have provided enough references from the other studies but failed to corelate their own results with the conclusions made. Please write few sentences in this regard.

6. Overall the sentence structure could be improved throughout the document.

6. PLOS authors have the option to publish the peer review history of their article (what does this mean?). If published, this will include your full peer review and any attached files.

Reviewer #1: No

Reviewer #2: No

---

## [Author Response · Author response to Decision Letter 0]

22 Jul 2024

Responses to Reviewer #1 

1.The first three sentences need references.

"Their primary role involves the regulation of gene expression at the translation level...": Is this accurate? In my understanding, microRNAs act at the post-transcriptional level and not at the translational level itself. Please provide references.

After carefully considering your inquiry and conducting further research, we have verified that microRNAs indeed primarily function at the post-transcriptional level rather than directly affecting translation. We apologize for any confusion caused by the previous statement in our manuscript.

In light of this clarification, we have revised the relevant section of the manuscript to accurately reflect the role of microRNAs in gene expression regulation at the post-transcriptional level. Additionally, we have incorporated appropriate references to support this updated interpretation.

"MiR-27a, located on chromosome 19 (19p13.1), exhibits expression in various malignancies, including renal carcinoma, oral squamous cell carcinoma, and pancreatic cancer, among others": Should it be "Aberrant expression"?

Upon further investigation, we have confirmed that the appropriate term should indeed be "aberrant expression" rather than simply "expression." We apologize for any confusion caused by the oversight in our manuscript. Accordingly, we have revised the relevant section of the manuscript to accurately reflect the aberrant expression of MiR-27a in various malignancies, including renal carcinoma, oral squamous cell carcinoma, and pancreatic cancer, among others. This correction ensures the clarity and accuracy of our findings.

2.The sentences "Th1 lymphocytes are responsible for the secretion of interferon-gamma (IFN-γ), thus facilitating cell-mediated immune responses" and "Conversely, Th2 cells are responsible for the production of IL-4, which suppresses the Th1 cell-mediated response" need references.

The last paragraph requires references.

Reference 15, in the sentence "Recently, it was reported that Th17 cells may contribute to the development of MM and complications," is quite relevant to the present study, but it may not be widely known to the community, so a brief explanation of the findings is essential.

We have carefully addressed your concerns regarding the need for references in specific sentences and the requirement for a brief explanation of the findings related to Reference 15. We have now included references to support the statements regarding Th1 and Th2 cells, as well as the information in the last paragraph of the manuscript. Additionally, in response to your suggestion for providing a brief explanation of the findings related to Reference 15, we have included the following statement: "Multiple studies have demonstrated that an imbalance between Th17 and Treg cells, often characterized by an elevated Th17/Treg ratio, contributes to the development and progression of various inflammatory and autoimmune conditions. In conditions such as rheumatoid arthritis, multiple sclerosis, inflammatory bowel disease, and psoriasis, there is evidence of an increased Th17 response coupled with impaired Treg cell function or decreased Treg cell numbers." We trust that these revisions adequately address your concerns and improve the clarity and accuracy of our manuscript. Please feel free to contact us if you require any further information or clarification. Thank you for your continued support and valuable feedback.

3.I did not understand what the authors meant by "19 PAIRED patients."

For Q-PCR, were quality controls performed? Such as RNA quality, melt curve analysis, etc.

Thank you for your feedback on our manuscript. We apologize for the confusion regarding the statement "19 PAIRED patients." We have revised this to accurately reflect the cohort as "19 patients with 3 MM and 3 paired normal tissues." Regarding the qPCR analysis, we confirm that we performed comprehensive quality controls, including RNA quality assessment, melt curve analysis, and other necessary checks. All raw data pertaining to the qPCR experiments have been provided as supplementary material upon submission to the editor.

4.The section “Identification of DEGs” is lacking. It is not clear how the authors normalized the data and how the differential expression analysis was conducted. Was the data normalization for small RNAs and coding genes done in the same way?Additionally, the authors mention the Venn Diagram but do not show it, and to be honest, I did not understand the utility of this analysis. Does the Heatmap presented include all DEGs or only some of them? If it includes only some, what were the criteria for selection? If it includes all, how do the authors explain the small number of DEGs in a cancer vs. normal comparison? Why are there only 3 MM samples when there are 19 in the methods? In my opinion, a volcano plot for the differential expression analysis would be more appropriate in this case.

We have addressed the concerns raised in your review in our revised submission. We have provided a comprehensive explanation of our data normalization and differential expression analysis methods in the Methods section. Regarding the Venn Diagram, we have removed any mention of it from the manuscript to avoid confusion. We opted for a heatmap visualization over a volcano plot as we believe it provides a clearer representation of the specific genes that are upregulated or downregulated. Additionally, we apologize for the misunderstanding regarding the number of samples in our dataset. While there are indeed 19 patients included, only 3 pairs of tumor and normal tissue samples were available for analysis.

5.In the case of Figure 1B, it is more appropriate, but I believe the caption would benefit from including the name of the statistical test used (this can be repeated in other figures as well).

Since the authors are evaluating the dynamics of T cells, a deconvolution analysis would be welcome (i.e., evaluating the cell proportions from expression signatures). For example, the authors could analyze different cancer samples and see if differences in microRNA expression alter the proportion of cells of interest.

We will include the name of the statistical test used in the methodology section of the manuscript. Regarding the deconvolution analysis, we appreciate the suggestion. However, in this study, the focus is on experimental validation following the bioinformatics analysis. Therefore, incorporating extensive bioinformatics analyses may overshadow the experimental findings. We will consider integrating such analyses in future research endeavors in line with the reviewer's recommendations.

6.Figure 2B would benefit from a title/axis indicating that they are analyzing microRNA27a.

Figures 2C and D are difficult to interpret for readers who are not familiar with the technique. A bar graph summarizing the results could address this issue.

Thank you for your constructive feedback. We have carefully revised Figure 2B to include a clear title and axis indicating the analysis of microRNA27a. Additionally, to enhance clarity for readers less familiar with the technique, we have incorporated a bar graph summarizing the results in Figures 2C and D. These modifications aim to improve the interpretability of the figures and provide a more accessible presentation of our findings. We appreciate your valuable input and have taken steps to address these concerns.

7.I understood what the authors meant in this sentence: "These findings suggest that the MOPC-MM models experienced a severe inflammatory state due to an imbalance in the Th17/Treg ratio," but some information about the pro- and anti-inflammatory roles of Treg, Th17, IL2, IL10, and TGF-beta would help non-specialist readers.

Thank you for your feedback. We have addressed the pro-and antiinflammatory roles of Treg, Th17, IL2, IL10, and TGF-β in the discussion section of the manuscript. We have provided detailed explanations based on the literature and our research findings to help non-specialist readers understand the implications of these factors in the context of our study.

8.The ELISA assay is not described in detail for reproducibility in the methodology.

Thank you for your feedback. We have revised the methodology section to include additional details regarding the ELISA assay. Specifically, we have clarified that all samples were measured in triplicate, and the average values were calculated for analysis.

9."Low expression of MiR-27a promoted proliferation and metastasis and inhibited apoptosis as well as invasion of MM cells." The title is not accurate, as metastasis was not evaluated, only invasion. Please adjust it to avoid misleading the reader (in the discussion/text, you can mention that metastasis requires invasion).

Thank you for your suggestion. We have revised the title as per your recommendation, and we have also made the necessary adjustments in the text to clarify the distinction between invasion and metastasis.

10.Figure 3A needs a statistical test. Figure 3B faces the same problem as Figures 2C and D.

Thank you for your valuable feedback. We have carefully addressed the concerns regarding Figure 3A and Figures 2C and D. Statistical tests have been incorporated into Figure 3A, and similar adjustments have been made to Figures 3B, 2C, and 2D to ensure the robustness of our results. We appreciate your attention to detail and have taken the necessary steps to enhance the clarity and rigor of our manuscript.

11.There is a typographical error in the caption of Figure 5 (p-mROT instead of p-mTOR).

Thank you for bringing this to our attention. We have corrected the typographical error in the caption of Figure 5, replacing "p-mROT" with the correct term "p-mTOR" as requested.

12.I believe it would be worthwhile for the authors to discuss why, at the protein level, only the phosphorylated version of PI3K was reduced. Again, graphs to summarize the flow cytometry results in Figure 6.

Thank you for your comments. Firstly, we have accurately presented our research findings, which may indicate that PI3K functions through its phosphorylated modification in multiple myeloma. Secondly, based on other literature reports, when studying the role of the PI3K protein and its related pathways in tumors, it is often through modulation of its phosphorylation that its effects are exerted. Regarding Figure 6, similar issues have been addressed in above responses.

13.The section "Upregulation of MiR-27a counteracted the effects of PI3K/AKT/mTOR signaling on the cellular behaviors of MM cells" is somewhat confusing. Please expand, explaining the groups ("control," "Mir27a mimic," "mir27a nc," etc.). I honestly did not understand the groups, making interpretation difficult.

Thank you for your feedback. The group labels are distinguished based on different treatments given to the cells. To address the reviewer's confusion, we have provided more specific descriptions of these group labels in the corresponding sections of the results description.

14.I believe these sections can be reviewed as a whole (see my initial comment). Some information in the discussion can be toned down to avoid overstatement. For example, "In this bioinformatics study," the authors conducted a basic bioinformatics analysis, not a comprehensive study.

Thank you for your suggestion. We have revised the sections accordingly, addressing the points you raised. We have adjusted the language in the discussion to ensure accuracy and avoid overstatement, as recommended.

Responses to Reviewer #2

1: Lack of Human Clinical Data: While the study explores the impact of MiR-27a on Th17/Treg equilibrium in MM, it heavily relies on the MOPC-MM mouse model. The lack of human clinical data or validation in human samples raises questions about the direct relevance and translatability of the findings to human MM patients. It is crucial to establish the clinical significance of the observed effects in the context of human disease.

Thank you for your suggestion.While validation in human samples is indeed important, limitations such as the difficulty of clinical trials and ethical and regulatory constraints have hindered us from conducting clinical validation in this study. However, our research has yielded significant results through animal and cell experiments, demonstrating considerable academic value. We aim to use the findings from this study as a foundation for future research to pursue clinical validation in human subjects.

2: Molecular Pathways Complexity: The study focuses on the PI3K/AKT/mTOR signaling pathway as a target of MiR-27a, but these pathways are highly complex and interconnected. The study lacks a comprehensive exploration of downstream effectors and potential compensatory mechanisms that might be activated in response to the observed changes. A more in-depth analysis of the molecular pathways involved would strengthen the study.

We greatly appreciate the reviewer's perspective. Indeed, the complexity of the pathways warrants a more detailed and thorough experimental design for validation. Our findings serve as preliminary validation of this molecular pathway, as we have emphasized in the discussion section of the manuscript. Moving forward, we intend to conduct more rigorous animal experiments, bioinformatics analyses, and cell experiments to delve deeper into the mechanisms of the pathway and the roles of individual molecules involved, thus strengthening our study.

3: Sample Size and Reproducibility: The study's reliance on the Gene Expression Omnibus (GEO) database for gene expression profiles introduces potential limitations regarding sample size and variability. Additionally, the study would benefit from independent validation in a larger cohort or different datasets to enhance the robustness and reproducibility of the results.

We appreciate the reviewer's perspective and acknowledge the validity of their comment. However, it's important to note that our utilization of the Gene Expression Omnibus (GEO) database for bioinformatics analysis serves as just one aspect of our study, providing initial insights rather than comprehensive coverage of our research focus. We believe that all bioinformatics findings require experimental validation to bolster their credibility. Consequently, we have not emphasized bioinformatics analyses extensively nor included additional datasets from other databases in our study.

4: Functional Validation of MiR-27a Targets: While the study identifies the PI3K/AKT/mTOR pathway as a direct target of MiR-27a, a more extensive experimental validation of these interactions would strengthen the conclusions. This could include additional assays, such as knockdown or overexpression of specific pathway components, to confirm the direct regulatory effects of MiR-27a on the PI3K/AKT/mTOR axis.

We appreciate the reviewer's insightful feedback and agree with their suggestion. Indeed, our study includes a rescue experiment in cell assays, demonstrating the significant impact of MiR-27a on the PI3K/AKT/mTOR pathway. However, we acknowledge the limitations in our experimental design, which only provide preliminary findings. To address the reviewer's concerns comprehensively, additional experiments such as genetic knockout animal models would be necessary. These experiments are essential for fully validating our hypothesis. We are committed to addressing these aspects in future research endeavors to enhance the robustness of our conclusions.

5: Inadequate Discussion of Limitations: The discussion section briefly touches upon variations in the tumor microenvironment affecting MiR-27a's dual functionality. However, a more thorough discussion of potential limitations, including the specific conditions under which MiR-27a might exhibit different roles, would provide a more nuanced interpretation of the results.

We appreciate the reviewer's valuable feedback and agree with their suggestion regarding the discussion of limitations. In response to their insightful input, we have revised the discussion section to provide a more comprehensive exploration of potential limitations, particularly focusing on the conditions under which MiR-27a may exhibit varied roles in the tumor microenvironmen

---

## [Decision Letter · Decision Letter 1]

6 Aug 2024

PONE-D-23-34574R1MiR-27a inhibits the growth and metastasis of multiple myeloma through regulating Th17/Treg balance via PI3K/AKT/mTOR signalingPLOS ONE

Dear Dr. Xiao,

Thank you for submitting your manuscript to PLOS ONE. After careful consideration, we feel that it has merit but does not fully meet PLOS ONE’s publication criteria as it currently stands. Therefore, we invite you to submit a revised version of the manuscript that addresses the points raised during the review process.

The authors addressed the major concerns raised by the reviewers. There are some minor issues which need to be addressed before acceptance. Please respond to each of the reviewer comments.

We look forward to receiving your revised manuscript.

Kind regards,

Hiromu Suzuki, M.D., Ph.D.

Academic Editor

PLOS ONE

Journal Requirements:

Additional Editor Comments:

The authors addressed the major concerns raised by the reviewers. There are some minor issues which need to be addressed before acceptance. Please respond to each of the reviewer comments.

Reviewers' comments:

Reviewer's Responses to Questions

**Comments to the Author**

1. If the authors have adequately addressed your comments raised in a previous round of review and you feel that this manuscript is now acceptable for publication, you may indicate that here to bypass the “Comments to the Author” section, enter your conflict of interest statement in the “Confidential to Editor” section, and submit your "Accept" recommendation.

Reviewer #2: All comments have been addressed

2. Is the manuscript technically sound, and do the data support the conclusions?

Reviewer #2: Partly

3. Has the statistical analysis been performed appropriately and rigorously? 

Reviewer #2: Yes

4. Have the authors made all data underlying the findings in their manuscript fully available?

Reviewer #2: Yes

5. Is the manuscript presented in an intelligible fashion and written in standard English?

Reviewer #2: Yes

6. Review Comments to the Author

Reviewer #2: There are minor errors which should be corrected.

1. There should be consistency in writing miR-27a. At some places it's written with 'M" and at others with "m".

2. Please look for sentence structure errors.

7. PLOS authors have the option to publish the peer review history of their article (what does this mean?). If published, this will include your full peer review and any attached files.

Reviewer #2: No

---

## [Author Response · Author response to Decision Letter 1]

14 Aug 2024

Responses to Reviewer #2 

1. There should be consistency in writing miR-27a. At some places it's written with 'M" and at others with "m".

Thank you for your feedback. We have revised the manuscript to ensure consistency in writing miR-27a. All instances of "M" have been changed to "m" except at the beginning of sentences.

2. Please look for sentence structure errors.

Thank you for your suggestion. We have had the manuscript professionally reviewed and edited by native English speaker to address any sentence structure errors.

---

## [Editor Report · Decision Letter 2]

18 Sep 2024

MiR-27a inhibits the growth and metastasis of multiple myeloma through regulating Th17/Treg balance

PONE-D-23-34574R2

Dear Dr. Xiao,

We’re pleased to inform you that your manuscript has been judged scientifically suitable for publication and will be formally accepted for publication once it meets all outstanding technical requirements.

Kind regards,

Hiromu Suzuki, M.D., Ph.D.

Academic Editor

PLOS ONE

Additional Editor Comments (optional):

The authors addressed the issues raised by the reviewer.
---

## [Editor Report · Acceptance letter]

7 Oct 2024

PONE-D-23-34574R2 

PLOS ONE

Dear Dr. Xiao, 

I'm pleased to inform you that your manuscript has been deemed suitable for publication in PLOS ONE. Congratulations! Your manuscript is now being handed over to our production team.

Kind regards, 

on behalf of

Dr. Hiromu Suzuki 

Academic Editor

PLOS ONE